# Targeting S-Nitrosylation to Overcome Therapeutic Resistance in NRAS-Driven Melanoma

**DOI:** 10.3390/cancers17122020

**Published:** 2025-06-17

**Authors:** Jyoti Srivastava, Sanjay Premi

**Affiliations:** Department of Tumor Microenvironment and Metastasis, Moffitt Cancer Center, 12902 USF Magnolia Drive, Tampa, FL 33612, USA

**Keywords:** NRAS-mutant melanoma, S-Nitrosylation, MEK-ERK signaling, targeted therapy, immune evasion, immunogenic cell death, redox modulation, tumor microenvironment, drug resistance

## Abstract

NRAS-mutant melanoma is difficult to treat due to resistance to current therapies like MEK inhibitors. This study highlights how a chemical process called S-nitrosylation helps tumor cells keep growing and avoid the immune system by modifying the function of key proteins like MEK and ERK. Blocking this process increases the efficacy of MEK inhibitors, boosts immune response, and helps immune cells better attack the tumor cells. These findings suggest a promising new strategy to improve treatment by combining redox-targeting drugs with existing therapies in NRAS-driven melanoma.

## 1. Introduction

The RAS family of small GTPases, comprising KRAS, NRAS, and HRAS isoforms, represents one of the most frequently mutated oncogenic drivers across human malignancies. These molecular switches regulate key signal transduction cascades, most notably the RAF-MEK-ERK (MAPK) and PI3K-AKT pathway, that govern cellular proliferation, differentiation, and survival. Oncogenic mutations in RAS genes, typically affecting codons 12, 13, and 61, result in constitutive activation by impairing intrinsic GTPase activity, thereby locking RAS proteins in a GTP-bound, active state. This aberrant signaling initiates and sustains tumorigenesis through dysregulated cell cycle progression, evasion of apoptosis, and enhanced metastatic potential. Among RAS-driven cancers, melanoma, a malignancy arising from melanocytes, exhibits a distinct mutational landscape in which NRAS mutations occur in approximately 15–20% of cases, predominantly at codon 61 [1]. NRAS-mutant melanomas demonstrate aggressive clinical behavior and pose significant therapeutic challenges, as effective targeted therapies remain elusive compared to their BRAF-mutant counterparts [2]. The intrinsic biochemical properties of RAS, including its picomolar affinity for guanine nucleotides and the absence of deep hydrophobic pockets amenable to small-molecule binding, have historically rendered it an “undruggable” target [3,4]. Consequently, therapeutic strategies have focused on inhibiting downstream effectors or exploiting synthetic lethal interactions, with limited clinical success to date.

The treatment landscape for melanoma has evolved dramatically with the advent of immune checkpoint inhibitors (ICIs) and targeted therapies. However, NRAS-mutant melanoma, a subtype comprising 15–25% of melanomas, remains refractory to most current treatments and is associated with a poorer prognosis compared to BRAF-mutant or wild-type variants [5,6]. Unlike BRAF-mutant melanomas, which respond to BRAF/MEK inhibition, MEK inhibition alone or in combination with immune checkpoint inhibitors, such as anti-PD-1 and anti-CTLA-4 antibodies, has provided limited clinical benefit in NRAS-mutant melanoma, highlighting the urgent need for novel therapeutic strategies. [7,8]. The recent study by Srivastava et al. [9] offers a compelling preclinical framework for overcoming this resistance by targeting nitric oxide (NO)-mediated S-nitrosylation, a reversible post-translational modification (PTM) that appears to regulate both signaling and immune evasion in NRAS-driven melanomas. This commentary aims to elucidate the current understanding of RAS-mediated oncogenic mechanisms in melanoma, role of S-nitrosylation in modulating such mechanisms, and the clinical potential of inhibiting S-nitrosylation to sensitize melanomas to targeted and immunotherapeutic approaches.

## 2. S-Nitrosylation: A Novel Regulator of MEK-ERK Signaling in NRAS-Mutant Melanoma

Protein S-nitrosylation is a reversible post-translational modification (PTM) involving the covalent attachment of a nitric oxide (NO) group to the thiol side chain of cysteine residues within target proteins. This modification modulates protein function, localization, stability, and interactions, thereby exerting broad regulatory control over numerous cell signaling pathways [10,11,12]. Mechanistically, S-nitrosylation can induce conformational changes in proteins that alter their enzymatic activity or binding affinity to other molecules. For example, S-nitrosylation of kinases or phosphatases can either activate or inhibit their catalytic functions, thereby modulating downstream phosphorylation cascades [13]. A classic instance is the S-nitrosylation of caspases, which inhibits their protease activity and thus regulates apoptosis [14]. Similarly, S-nitrosylation of the transcription factor NF-κB can affect its DNA binding ability and transcriptional activity, influencing inflammatory responses [15].

S-nitrosylation also impacts receptor signaling at the plasma membrane. For example, S-nitrosylation of the epidermal growth factor receptor (EGFR) can modulate its kinase activity and downstream MAPK signaling [16]. In the context of oncogenic signaling, S-nitrosylation of proteins within the RAS-MEK-ERK pathway has been shown to enhance or sustain signaling activity, contributing to tumor progression [17]. This occurs through direct modification of pathway components, altering their stability or interaction networks. The dynamic and reversible nature of S-nitrosylation enables cells to finely tune signaling responses in accordance with fluctuating nitric oxide levels, which are often elevated during inflammation, hypoxia, or oncogenic stress [12,18]. Enzymatic denitrosylation systems, such as thioredoxin and S-nitrosoglutathione reductase, help maintain S-nitrosylation homeostasis by removing NO groups, ensuring signaling fidelity [19].

In summary, S-nitrosylation is a versatile regulator of cell signaling, acting through modification of key signaling proteins to modulate cellular processes including proliferation, apoptosis, immune responses, and differentiation. Dysregulation of S-nitrosylation is implicated in various diseases, including cancer, where it can promote aberrant signaling and resistance to therapy.

The MAPK pathway, which includes RAS, RAF, MEK, and ERK, is a central driver of melanoma proliferation and survival [20,21]. In NRAS-mutant tumors, constitutive activation of this pathway is a hallmark of oncogenic signaling. Srivastava et al. reveal that this hyperactivation is potentiated by NO-induced S-nitrosylation of core MAPK components, including NRAS, MEK, ERK, and downstream effectors like RSK1 and DUSPs [9]. Importantly, they demonstrate that S-nitrosylation promotes phosphorylation of MEK and ERK, thereby stabilizing and reinforcing MAPK signaling. These observations support earlier findings that S-nitrosylation can modulate kinase activity through conformational shifts or changes in subcellular localization [10,11]. 

A particularly innovative aspect of the study is the identification of key S-nitrosylation sites on ERK, specifically C178 and C233. Site-directed mutagenesis at these residues impaired ERK phosphorylation and suppressed melanoma cell proliferation, providing functional evidence for their regulatory role. This finding diverges from prior studies where S-nitrosylation of extracellular signal-regulated kinase (ERK) at cysteine residues has been shown to attenuate kinase activity, suppress downstream signaling, and induce apoptosis in cancer models. Prior studies have identified C183 as a critical S-nitrosylation site on ERK1, with modification at this residue leading to reduced ERK phosphorylation and inhibition of cell proliferation in non-melanocytic cancer cell lines such as glioma and breast cancer cells [22,23]. Mutation of C183 to alanine (C183A) prevents this inhibitory modification, thereby restoring ERK activation and enhancing cellular resistance to apoptosis.

While these findings provide compelling evidence of ERK S-nitrosylation as a tumor-suppressive mechanism, they were primarily conducted in non-melanoma contexts and may not fully represent signaling dynamics in melanoma. In our recent work [9], we extended this understanding through mass spectrometry-based S-nitrosylome profiling coupled with chemical and genetic modulation of nitric oxide synthase (NOS). Our study revealed that in NRAS-mutant melanoma cells, ERK remains persistently activated despite MEK inhibition, a phenomenon driven by S-nitrosylation. This persistent activation undermines the efficacy of MEK inhibitors (MEKi) as monotherapies. Crucially, we demonstrated that co-treatment with MEKi and NOS inhibitors (NOSi) led to sustained suppression of ERK-MAPK signaling and markedly enhanced sensitivity of tumor cells to MEKi. These findings establish S-nitrosylation as a non-genetic driver of therapeutic resistance in melanoma and suggest that NOS activity modulates ERK signaling in a context-specific manner [9].

Moreover, our observations highlight potential methodological and biological disparities between studies. While prior reports primarily focused on C183 and employed a single NO donor in non-melanoma cells [22,23], our work utilized a broader range of chemical sources and orthogonal validation strategies, including ERK mutants C178 and C233, across multiple cell types, including melanocytic lineages. These differences may account for contrasting conclusions regarding the functional outcomes of ERK S-nitrosylation and point to the necessity of considering cell-type specificity in S-nitrosylation research.

## 3. Immune Reprogramming Through Inhibition of S-Nitrosylation

Protein S- is a crucial regulator of immune signaling pathways, influencing key proteins involved in T cell activation, apoptosis, and cytokine signaling [10,12]. In the tumor microenvironment, aberrant S-nitrosylation can modulate immune checkpoint molecules and suppress antitumor immune responses, facilitating tumor immune evasion [24]. By altering redox-sensitive signaling cascades, S-nitrosylation dynamically regulates the balance between pro- and anti-inflammatory states, impacting immune cell function and therapeutic responses. Thus, understanding S-nitrosylation-mediated immune regulation offers new opportunities to enhance cancer immunotherapy strategies.

One of the most striking outcomes of S-nitrosylation inhibition in our investigations was its ability to convert non-immunogenic cell death into immunogenic cell death (ICD), thereby restoring anti-tumor immunity. ICD is characterized by the release of damage-associated molecular patterns (DAMPs) such as calreticulin, HMGB1, and phosphorylated eIF2α, which act as immunologic adjuvants that promote dendritic cell maturation and T-cell priming [25]. Srivastava et al. show that inhibition of S-nitrosylation not only facilitates DAMP release but also prevents their inactivation via S-nitrosylation, suggesting that NO signaling acts as a suppressor of ICD in melanoma [9].

These results are in line with earlier studies indicating that NOS activity can dampen immune responses in the tumor microenvironment [26]. Moreover, the combination of NOSi and MEKi induced robust infiltration of CD8+ T cells, dendritic cells, and macrophages in vivo, which was corroborated in ex vivo dendritic cell–T cell co-cultures [9]. This finding is significant, as MEK inhibition has been previously associated with enhanced tumor immunogenicity [27], but the molecular underpinnings remained unclear.

Srivastava et al. extend this paradigm by positioning S-nitrosylation as a negative regulator of tumor immunogenicity. Our data also suggests a dual mechanism for S-nitrosylation inhibition: one involving direct attenuation of oncogenic signaling and another reprogramming the tumor microenvironment to restore immune surveillance. Both mechanisms are summarized below in A and B and in Figure 1:


**A.** 
**MEKi Signaling Inhibition via S-nitrosylation Modulation**
NOS-mediated S-nitrosylation constitutively activates MEK-ERK signaling in NRAS-mutant melanoma.S-nitrosylation occurs on key proteins including MEK, ERK, NRAS, DUSPs, and RSK1, enhancing MAPK pathway activity.NOS inhibition reduces S-nitrosylation, allowing recycling of non-nitrosylated proteins.De-nitrosylation sensitizes melanoma cells to MEK inhibitors by overcoming S-nitrosylation-driven MEKi resistance.NOS inhibition and MEK inhibition together induce apoptosis in melanoma cells.NOS inhibition alone has minimal effect unless prolonged; main impact is synergistic with MEK inhibition.

**B.** 
**Immune Response Reactivation through S-nitrosylation Inhibition and MEKi**
NOS inhibition combined with MEKi induces apoptosis, releasing de-nitrosylated DAMPs.These de-nitrosylated DAMPs are highly immunogenic and induce immunogenic cell death (ICD).ICD triggers release of damage-associated molecular patterns (DAMPs) that activate dendritic cells (DCs).Activated DCs prime and proliferate CD8+ T cells, enhancing cytotoxic anti-tumor immunity.NOS inhibition promotes macrophage infiltration, supporting tumor phagocytosis and immune activation.S-nitrosylation modulates CD47-CXCR4 interaction affecting tumor immune evasion and “immune surrender” mechanisms.Combined effect results in restricted tumor growth and reversal of immune evasion.



Based on these two mechanisms, we recommend sequential therapy with NOS inhibition followed by MEK inhibition to induce tumor cell apoptosis and reactivate anti-tumor immunity in melanoma.

## 4. Implications for Macrophage-Mediated Immunity and “Immune Surrender”

Beyond dendritic cell and T cell activation, NOSi treatment enhanced macrophage infiltration into tumors. This is particularly notable given emerging evidence that tumor-associated macrophages (TAMs) can directly phagocytose viable cancer cells via modulation of the CD47/CXCR4/CXCL12 axis, a process referred to as “immune surrender” [28].

Immune surrender is a mechanism of immune surveillance distinct from immunogenic cell death, whereby stressed or damaged cells actively facilitate their recognition and elimination by the immune system without undergoing cell death themselves. Unlike immunogenic cell death, which releases danger signals that recruit immune cells, immune surrender involves the regulated exposure of “eat-me” signals on the cell surface, promoting phagocytosis and immune clearance. This process enables the immune system to eliminate potentially harmful cells while preserving tissue integrity. Immune surrender thus represents a complementary strategy for maintaining homeostasis and preventing tumor development.

In NRAS-mutant melanoma, S-nitrosylation emerges as a critical regulator of this process. Srivastava et al. indicate that inhibiting S-nitrosylation enhances macrophage infiltration and tumor phagocytosis, thereby counteracting immune evasion. Specifically, S-nitrosylation appears to modulate CD47 internalization via its interaction with CXCR4, suggesting that nitric oxide-dependent post-translational modifications influence the balance between tumor cell survival and immune clearance. This interplay between S-nitrosylation and immune surrender complements immunogenic cell death pathways and offers novel therapeutic opportunities to boost antitumor immunity by promoting macrophage-mediated elimination of live tumor cells. Understanding how S-nitrosylation governs immune surrender will be essential for designing combinatorial therapies that overcome resistance in aggressive NRAS-mutant melanomas.

## 5. Translational and Therapeutic Potential

Therapeutic intervention with nitric oxide synthase (NOS) inhibitors offers a promising strategy to circumvent this resistance. By blocking NOS activity, NOS inhibitors reduce S-nitrosylation levels on critical signaling proteins, thereby destabilizing ERK phosphorylation and amplifying the suppressive effects of MEK inhibitors on tumor growth. This combination demonstrates superior efficacy in preclinical models, suggesting that NOS inhibition can sensitize NRAS-mutant melanoma cells to targeted therapy.

Beyond tumor cell-intrinsic effects, S-nitrosylation inhibition profoundly modulates the tumor microenvironment. It enhances the release and function of immunostimulatory molecules, transforming tumor cell death into an immunogenic form. This shift facilitates the activation and maturation of dendritic cells, increases infiltration of CD8+ cytotoxic T lymphocytes, and promotes a pro-inflammatory milieu conducive to tumor eradication. Moreover, S-nitrosylation blockade stimulates macrophage recruitment and promotes macrophage-mediated clearance of tumor cells by modulating “eat-me” signals through pathways involving CD47 and CXCR4. This supports a concept known as immune surrender, whereby tumor cells are targeted for destruction by innate immune mechanisms independently of cell death signaling.

Taken together, targeting S-nitrosylation represents a multi-pronged therapeutic approach that integrates suppression of oncogenic MAPK signaling with enhanced immune-mediated tumor clearance. This strategy holds significant potential to improve outcomes in NRAS-driven melanoma by overcoming resistance and engaging both adaptive and innate immunity. NOS inhibitors such as L-NAME and 1400 W have been tested in various preclinical models and may be repurposed for combination therapy in melanoma. Given that global NOS inhibition could have systemic effects, isoform-specific or tumor-targeted NOSi approaches may be needed to ensure safety. Srivastava et al. showed that inhibiting individual NOS isoforms produced similar outcomes in reducing S-nitrosylation and sensitizing melanoma to MEK inhibition. However, since inhibiting one isoform is often compensated by heightened activity of other NOS isoforms, further analysis is needed to identify which specific isoforms are essential for S-nitrosylation-mediated MEKi resistance and immune evasion. Accordingly, new models are needed where only one NOS isoform is active at a time to precisely determine their individual roles.

Alternatively, direct inhibitors of S-nitrosylation, or molecules that stabilize de-nitrosylated forms of ERK and MEK, could be developed. Moreover, this study provides a rationale for using S-nitrosylation signatures as biomarkers of MEKi resistance or immune evasion. Profiling the tumor nitrosylome could enable patient stratification and identify individuals most likely to benefit from NOSi + MEKi therapy. Finally, these insights could extend beyond melanoma. S-nitrosylation has been implicated in KRAS-driven colorectal and pancreatic cancers [29,30], and several other cancer types [31,32], suggesting broader applications of this therapeutic concept in RAS-driven and other malignancies.

In normal cells, KRAS is S-nitrosylated primarily at cysteine 118 (Cys118), which enhances its GDP/GTP exchange and promotes activation [33]. KRAS mutations such as G12C and G13C can apparently generate additional S-nitrosylation sites, further increasing GTP binding and oncogenic activity [33]. Furthermore, nitrosylated KRAS positively regulates inducible nitric oxide synthase (iNOS), creating a feedback loop that sustains NO production and S-nitrosylation levels [34,35].

Similar to KRAS, wild-type NRAS undergoes S-nitrosylation at Cys118, supporting proliferative signaling [17]. In NRAS-mutant melanoma, S-nitrosylation contributes to MEK-ERK pathway hyperactivation and resistance to MEK inhibitors [9]. Inhibition of S-nitrosylation, combined with MEK inhibitors, has shown significant tumor growth suppression in mouse models by promoting immunogenic cell death (ICD) and reactivating anti-tumor immunity [9].

In summary, both KRAS and NRAS can be nitrosylated, though the S-nitrosylation sites, and S-nitrosylation mediated activation might vary. Mutant KRAS harbors additional S-nitrosylation sites, potentially amplifying oncogenic signaling. Also, NRAS-specific S-nitrosylation appears critical for MEKi resistance and immune evasion in melanoma, making it a promising therapeutic target. Thus, S-nitrosylation significantly impacts the function of both KRAS and NRAS. KRAS mutations may introduce further S-nitrosylation sites, intensifying oncogenic potential. Importantly, targeting S-nitrosylation in NRAS-mutant melanoma holds therapeutic promise by reversing drug resistance and stimulating anti-tumor immune responses. However, the effects of S-nitrosylation are context-dependent, varying with the Ras isoform, mutation type, and tumor environment.

## 6. Conclusions and Future Directions

Protein S-nitrosylation emerges as a dynamic and reversible post-translational modification that critically regulates both oncogenic signaling and immune modulation in NRAS-mutant melanoma. By covalently modifying cysteine residues on key signaling proteins such as MEK and ERK, S-nitrosylation sustains aberrant MAPK pathway activation and promotes melanoma cell proliferation and survival, even in the presence of targeted therapies. Beyond its role in tumor cell signaling, S-nitrosylation exerts potent immunosuppressive effects within the tumor microenvironment. It impairs immunogenic cell death (ICD), alters the release and recognition of danger-associated molecular patterns (DAMPs), and modulates immune checkpoint pathways—thereby facilitating immune evasion and resistance to both MAPK-targeted and immune checkpoint therapies.

Pharmacological inhibition of nitric oxide synthase (NOS) disrupts these tumor-promoting mechanisms by attenuating sustained ERK phosphorylation and restoring the immunogenicity of tumor cell death. This, in turn, promotes dendritic cell activation, CD8+ T cell infiltration, and macrophage-mediated phagocytosis of tumor cells via immune surrender pathways. These data underscore the dual role of S-nitrosylation as both a driver of oncogenic signaling and a suppressor of anti-tumor immunity, identifying it as a critical, non-genetic contributor to therapeutic resistance in NRAS-mutant melanoma.

Targeting redox regulation through modulation of S-nitrosylation represents a promising strategy to overcome the multifaceted therapeutic challenges posed by this melanoma subtype. However, effective translation requires a series of defined next steps. First, refined preclinical models with conditional regulation of individual NOS isoforms are essential to dissect isoform-specific roles, particularly given compensatory upregulation observed when a single isoform is inhibited. These models will clarify which NOS isoforms are essential for sustaining S-nitrosylation-driven resistance and immune suppression. Validation of these mechanisms in advanced preclinical systems—including patient-derived xenografts, syngeneic models, and humanized immune platforms—will be vital for capturing the complexity of tumor–immune interactions. Concurrently, pharmacologic optimization of NOS inhibitors to enhance isoform selectivity and reduce off-target effects is critical. The development of predictive biomarkers, such as nitrosylated MAPK components or DAMP release profiles, will facilitate clinical translation by informing patient selection and response monitoring.

Finally, early-phase clinical trials evaluating the sequential or combinatorial use of NOS and MEK inhibitors in NRAS-mutant melanoma are warranted. These studies should prioritize pharmacodynamic endpoints, immune reactivation markers, and preliminary efficacy assessments. Given the conserved nature of S-nitrosylation across RAS-driven signaling and immune escape pathways, this approach may also extend to other malignancies characterized by redox dysregulation and MAPK pathway dependence.

In summary, the therapeutic modulation of protein S-nitrosylation offers a compelling new avenue to overcome resistance and reinvigorate anti-tumor immunity in NRAS-mutant melanoma, warranting further preclinical and clinical investigation.

## Figures and Tables

**Figure 1 cancers-17-02020-f001:**
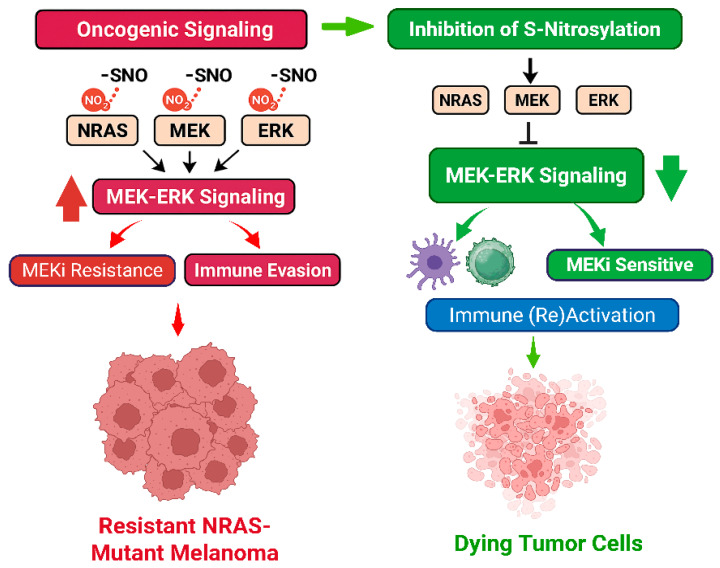
Denitrosylation Sensitizes Melanoma to MEK Inhibition by Modulating ERK-MAPK Activity and Promoting Immune Reactivation: In the melanoma tumor microenvironment (TME), elevated nitric oxide synthase (NOS) activity causes sustained S-nitrosylation of ERK-MAPK pathway proteins, NRAS, MEK, and ERK, driving pathway hyperactivation and resistance to MEK inhibitors (MEKi). Denitrosylation, either through NOS inhibition, nitric oxide scavenging, or site directed mutagenesis of the targeted cysteines, restores normal regulation of these proteins and resensitizes melanoma cells to MEKi. Immunogenic signals from melanoma cells dying in response to MEKi promote immune reactivation against the tumor. This response contributes to the clearance of residual tumors and remaining melanoma cells. Supported by recent findings by Srivastava et al. [9].

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
