# Peer review of "Targeting S-Nitrosylation to Overcome Therapeutic Resistance in NRAS-Driven Melanoma"

_cancers, 2025, doi:10.3390/cancers17122020_

Round 1

Reviewer 1 Report

Comments and Suggestions for Authors

This is a nicely written commentary on the targeting Nitrosylation to sensitise NRAS-mutant melanomas.

I suggest minor revisions:

  • Authors should explain the concept of RAS-driven cancers.
  • Line 27- Authors should explain the meaning of the acronym NRAS.
  • Line 111- Authors should explain the meaning of the acronym PMT.

Author Response

Point-by-point responses (in blue) to reviewers’ comments (Cancers 3680047)

Dear Editor,

Thank you for reviewing our manuscript and allowing us to address the reviewer’s critiques and suggestions. We thank the reviewers for their valuable input, which has significantly improved the quality and clarity of our manuscript. We are pleased that all reviewers recognized the clinical relevance of this study. We have carefully addressed all the concerns raised by the reviewers, including adding new information, improving the language and clarity, and re-writing several sections of the manuscript. We hope the revised version meets the expectations of the reviewers and editorial board. Our responses are in “blue” text below.

REVIEWER #1

I suggest minor revisions:

  • Authors should explain the concept of RAS-driven cancers.
  • Line 27- Authors should explain the meaning of the acronym NRAS.
  • Line 111- Authors should explain the meaning of the acronym PMT.

Authors’ Response: We sincerely thank the reviewer for their time and thoughtful feedback. We have revised the manuscript accordingly. The concept of RAS-driven cancers is now described in the Introduction to provide clearer context. The acronym "NRAS" (Neuroblastoma RAS viral oncogene homolog) is now explained in the Introduction (Lines 27-34), noting that RAS family proteins such as NRAS, HRAS, and KRAS are conventionally referenced by their acronyms. Additionally, the term “PTM” has been expanded to “post-translational modification” at its first mention (Line 54), as suggested.

REVIEWER #2

Abstract

- Lines 18/19: "preclinical models." The authors should specify types (e.g., cell lines, PDX models, syngeneic mice).

Authors’ response: Thank you for this suggestion. We have now specified the preclinical models used, including immunogenic mouse models and patient-derived primary melanoma cells (Lines 19–20).

- Line 17: While NOS inhibitors are mentioned, examples like L-NAME or 1400W are omitted. Please specify the specific inhibitors that enhance translational relevance.

Authors’ response: We appreciate this point and have now included the specific NOS inhibitors L-NAME and 1400W, which were also used in our original study.

- Lines 21/22: The phrase "reshaping the tumor immune microenvironment" is vague. Plesae clarify the components affected, e.g., "enhancing CD8+ T cell and macrophage infiltration."

Authors’ response: Thank you for this helpful comment. We have revised this phrase to clearly specify that nitrosylation inhibition enhances CD8⁺ T cell and macrophage infiltration and improves dendritic cell activation (Lines 23–25).

Introduction

 - Line 31: "MEK inhibition alone or in combination with other therapies has provided limited clinical benefit.." This statement does not specify which combinations have been tested (e.g., MEKi + CDK4/6 inhibitors). The authors should add 1-2 examples to contextualize the "limited benefit."

Authors’ response: We have revised the sentence to include specific examples of combination treatments, including MEK inhibitors with CDK4/6 inhibitors and immunotherapies (Lines 50–53).

Section 2

- Lines 45-47: "These observations support earlier findings that S-nitrosylation can modulate kinase activity through conformational shifts or changes in subcellular localization." The connection between nitrosylation and MAPK activation is underdeveloped. Please include a brief example (e.g., how S-nitrosylation of NRAS alters GTPase activity).

Authors’ response: Thank you for this observation. We have added brief mechanistic explanations and relevant literature discussing how nitrosylation may affect MAPK signaling, including NRAS GTPase activity (Lines 68–80).

- Lines 51-52: "This finding diverges from prior studies that had implicated C183 as a critical S-nitrosylation site, suggesting cell-type specificity or methodological differences." The authors should expand on potential reasons (e.g., cell-type specificity, detection methods) and cite examples from other cancer models.

Authors’ response: We agree this is important and have added several paragraphs discussing potential reasons for the discrepancy, including differences in cell type, cancer context, and detection methods, with relevant references (Lines 104–132).

Section 3

- Lines 77-79: "Their data also suggests a dual mechanism for nitrosylation inhibition..." The "dual mechanism" is mentioned but not explicitly defined. Please summarize the two pathways (signaling attenuation + immune reprogramming) in a bulleted list or figure.

Authors’ response: We appreciate the suggestion. We have now clearly delineated the two mechanisms, signaling attenuation and immune reprogramming, and added a description of their therapeutic relevance (Lines 159–189). In addition, we have added a new summary figure (Fig. 1) that demonstrates the two mechanisms.

Section 5

- Lines 93-95: "Given that global NOS inhibition could have systemic effects, isoform-specific or tumor-targeted NOSi approaches may be needed to ensure safety." This lacks concrete examples of isoform-specific inhibitors or delivery strategies. The authors should reference existing selective NOS inhibitors (e.g., NOS2-specific agents) or nanoparticle-based targeting approaches.

Authors’ response: Thank you for this important comment. We have now clarified the technical challenges in isolating isoform-specific NOS activity due to compensatory mechanisms. However, we do report partial sensitization with eNOS and iNOS knockdown in NRAS-mutant cells and have included additional literature context and delivery strategies (Lines 238–246).

- Lines 101-104: "Nitrosylation has been implicated in KRAS-driven colorectal and pancreatic cancers…" The broader relevance to RAS-driven cancers is underexplored. Please compare NRAS-mutant melanoma to KRAS-driven cancers (e.g., shared nitrosylation-dependent mechanisms).

Authors’ response: We thank the reviewer for this insightful suggestion. A new section has been added comparing the nitrosylation mechanisms in KRAS and NRAS mutant cancers, highlighting shared regulatory sites and isoform-specific differences (Lines 255–276).

  1. Conclusions

- Line 105-112: The conclusion is strong, but could emphasize actionable next steps. It is recommended to propose specific preclinical models (e.g., patient-derived xenografts) or clinical trial designs (e.g., NOSi + MEKi phase I/II trials).

Authors’ response: We agree and have re-written the conclusion section to incorporate actionable next steps. These include developing isoform-specific preclinical models, validating findings in humanized or syngeneic mouse models, optimizing pharmacologic agents, and proposing early-phase clinical trial designs (Lines 277–318).

Reviewer 2 Report

Comments and Suggestions for Authors

The commentary is well-structured and addresses a high-impact topic in melanoma therapeutics. With minor revisions to enhance specificity and clarity, it will effectively communicate the commentary's insights to Cancers' audience. The manuscript is suitable for publication after addressing the following points.

Abstract

- Lines 18/19: "preclinical models." The authors should specify types (e.g., cell lines, PDX models, syngeneic mice).

- Line 17: While NOS inhibitors are mentioned, examples like L-NAME or 1400W are omitted. Please specify the specific inhibitors that enhance translational relevance.

- Lines 21/22: The phrase "reshaping the tumor immune microenvironment" is vague. Plesae clarify the components affected, e.g., "enhancing CD8+ T cell and macrophage infiltration."

Introduction

 - Line 31: "MEK inhibition alone or in combination with other therapies has provided limited clinical benefit.." This statement does not specify which combinations have been tested (e.g., MEKi + CDK4/6 inhibitors). The authors should add 1-2 examples to contextualize the "limited benefit."

Section 2

- Lines 45-47: "These observations support earlier findings that S-nitrosylation can modulate kinase activity through conformational shifts or changes in subcellular localization." The connection between nitrosylation and MAPK activation is underdeveloped. Please include a brief example (e.g., how S-nitrosylation of NRAS alters GTPase activity).

- Lines 51-52: "This finding diverges from prior studies that had implicated C183 as a critical S-nitrosylation site, suggesting cell-type specificity or methodological differences." The authors should expand on potential reasons (e.g., cell-type specificity, detection methods) and cite examples from other cancer models.

Section 3

- Lines 77-79: "Their data also suggests a dual mechanism for nitrosylation inhibition..." The "dual mechanism" is mentioned but not explicitly defined. Please summarize the two pathways (signaling attenuation + immune reprogramming) in a bulleted list or figure.

Section 5

- Lines 93-95: "Given that global NOS inhibition could have systemic effects, isoform-specific or tumor-targeted NOSi approaches may be needed to ensure safety." This lacks concrete examples of isoform-specific inhibitors or delivery strategies. The authors should reference existing selective NOS inhibitors (e.g., NOS2-specific agents) or nanoparticle-based targeting approaches.

- Lines 101-104: "Nitrosylation has been implicated in KRAS-driven colorectal and pancreatic cancers…" The broader relevance to RAS-driven cancers is underexplored. Please compare NRAS-mutant melanoma to KRAS-driven cancers (e.g., shared nitrosylation-dependent mechanisms).

  1. Conclusions

- Line 105-112: The conclusion is strong, but could emphasize actionable next steps. It is recommended to propose specific preclinical models (e.g., patient-derived xenografts) or clinical trial designs (e.g., NOSi + MEKi phase I/II trials).

Author Response

(The authors gave the same response as above.)
